# Diet Quality Variation among Polish Older Adults: Association with Selected Metabolic Diseases, Demographic Characteristics and Socioeconomic Status

**DOI:** 10.3390/ijerph20042878

**Published:** 2023-02-07

**Authors:** Robert Gajda, Ewa Raczkowska, Małgorzata Sobieszczańska, Łukasz Noculak, Małgorzata Szymala-Pędzik, Michaela Godyla-Jabłoński

**Affiliations:** 1Department of Human Nutrition, Faculty of Biotechnology and Food Sciences, Wrocław University of Environmental and Life Sciences, Chełmońskiego 37, 51-630 Wroclaw, Poland; 2Clinic Department of Geriatrics, Faculty of Medicines, Wrocław Medical University, M. Curie-Skłodowskiej 66, 50-369 Wroclaw, Poland

**Keywords:** metabolic diseases, diet quality, demographic characteristics, socioeconomic status, older adults

## Abstract

A lot of civilization diseases are related to a low-quality diet, which is often determined by environmental factors. The aim of the present study was to assess the relationship between the quality of diet and the selected metabolic diseases, as well as demographic characteristics and socioeconomic status among Polish seniors. The study was conducted on the basis of the KomPAN questionnaire (Questionnaire for Dietary Views and Habits). The research sample was chosen arbitrarily. In addition, in order to diversify the research sample, the use of the snowball method was used. The study was conducted from June to September 2019 in a group of 437 people aged 60 or more years in two regions of Poland. Two diet quality indices with a potentially beneficial (pHDI-10) and adverse impact on health (pHDI-14) were selected based on data on the frequency of consumption of 24 food groups using the KomPAN questionnaire data development procedure. Based on the intensities (low, moderate, high) and combinations of these indices, three diet quality index profiles were developed with potentially different influences on health: lower (lowest), middle (intermediate) and upper (highest). Logistic regression was used to evaluate the relationship between diet quality indices, some metabolic diseases (obesity, arterial hypertension, diabetes type 2), demographic characteristics (gender, age, place of residence), and socioeconomic status (low, moderate, high). It was shown that in the examined seniors with selected metabolic diseases, the higher quality diet was more common among women, urban inhabitants and subjects with higher socioeconomic status. In turn, among the elderly with obesity, a high-quality diet was observed more often in people aged 60–74 years and those with type II diabetes at ages 75 years or more. The relationships between diet quality, demographic characteristics and socioeconomic status were demonstrated, but it was not possible to obtain unambiguous results on the relationship of these variables with the occurrence of metabolic diseases. Further extended studies should assess the importance of diet quality in reducing the risk of metabolic diseases in the elderly, taking into account the variability resulting from the environmental characteristics of the study population.

## 1. Introduction

Forecasts of the increase in the world’s elderly population also apply to Poland [1]. Currently, in Poland, people aged 80 and over account for 18% of the population, while by 2035, the size of this subpopulation will be doubled; this phenomenon is named “double aging” [2]. According to Rosset, the marker for exceeding the threshold of demographic old age is a greater than 12% share of the population older than 60 years of age [3]. In the Polish nation, the subpopulation over 60 years was 20.2% in 2011, which means that the aging process is very advanced [4]. Therefore, both the healthcare system and individual people should become more aware of the health problems caused by the aging process, as well as the importance of good health and high quality of life in the later stages of life [5,6]. Several health problems arise during aging, including chronic diseases [5,7] and malnutrition [5,8]. Among chronic diseases, obesity, arterial hypertension, atherosclerotic cardiovascular disease and type 2 diabetes are commonly observed [9,10]. In 2019 in Poland, as many as two-thirds of people (66.3%) aged 60 years or more reported long-term health problems or chronic diseases [11].

The issue of diet quality has received a lot of attention in much of the nutritional research. Despite its widespread use, the concept of diet quality is poorly defined and difficult to measure. No consensus can be reached on how to universally define diet quality and develop a standard indicator for assessing it with reliable predictive properties [12]. Numerous nutritional indicators were developed, tested and validated to reflect various aspects of dietary qualities [13]. In recent years, the interest in using specific indices to assess diet quality and their impact on non-communicable diseases has increased. These indices are used mainly in developed countries and were developed for research among Americans, e.g., the HEI (Healthy Eating Index) and DQI (Diet Quality Index) [14,15,16]; Northern European residents, such as the MDS (Mediterranean Diet Score) [17]; and for of the Chinese population, namely, the DBI (Dietary Balance Index) [18]. In Poland, based on the available literature [19,20,21], in order to comprehensively assess the quality of a diet, two indicators were developed. One of them brings together foods with potential health benefits, namely, the pHDI (Pro-Healthy Diet Index), and the second brings together foods unfavorable to health, namely, the nHDI (Non-Healthy Diet Index) [22]. 

Older adults are at serious risk of nutritional errors as a consequence of food deficiencies and/or low-quality diets [23,24], psycho-social and economic problems [25], and involuntary changes in the body and general health state [5,26]. An improper diet is associated with deteriorating health [5,27] and quality of life [28,29]. Among the elderly, a low-quality diet is usually the result of numerous nutritional errors, including too low consumption of vegetables, fruits, legumes, wholegrain cereal products, fish, water and other liquids, as well as excessive intake of sugar and sweets, fatty meat and cured meats [5,27]. 

Unfortunately, food security among the elderly and, therefore, the quality of their diet is limited by their health status [5], physical functioning and activities [30], interpersonal relations and social support [24,31], as well as the distance from the place of residence to the grocery store [23,32]. It was also shown that the risk of health disorders is not only determined by the quality of the diet but also by demographic factors, such as gender, age, place of residence [26] and socioeconomic status [33,34].

For the purpose of improving the quality of a diet, the Mediterranean diet (Med) and Dietary Approaches to Stop Hypertension (DASH) are recommended, especially in European countries [35,36]. These diets are closely related to an increased life expectancy [36] and lower risk of metabolic diseases [37,38]. In turn, the MIND diet, which is a hybrid diet of Med and DASH, shows a protective effect against cognitive dysfunctions [39].

The relationship between a high-quality diet and a lower risk of metabolic diseases, especially type 2 diabetes [36], cardiovascular disease [40,41,42], and overweight and obesity, is relatively well explained in the literature [43]. Research results are also available showing that there is a relationship between selected sociodemographic characteristics of the elderly and the quality of their diet. Available data include the place of residence, gender [44] and socioeconomic status [45]. However, the relationships between the quality of a diet; the occurrence of metabolic diseases; and demographic, economic and psycho-social characteristics are complex and ambiguous. In addition, to our knowledge, the available literature lacks papers that treat the problem of linking diet quality to the incidence of specific metabolic diseases in such a broad way while also taking into account demographic characteristics and socioeconomic status. There is also a lack of data on Polish seniors. The present study was based on the assumption that metabolic disease occurrence is associated with a low-quality diet, although this relationship is different for various metabolic conditions and depends on demographic characteristics and socioeconomic status of the population examined. In this context, the aim of the study was to assess the relationship between diet quality, selected metabolic diseases, demographic characteristics and socioeconomic status in a sample of Polish seniors. 

## 2. Materials and Methods 

### 2.1. Study Design and Sample

This paper is the result of a scientific study conducted under the title “Dietary habits of the elderly and their selected determinants”. The study was conducted on the basis of the KomPAN questionnaire (Questionnaire for Dietary Views and Habits), which was developed by the Committee on Human Nutrition Science of the Polish Academy of Sciences. The study was not funded by any means. The KomPAN questionnaire contains four separate sections with thematically grouped questions on eating habits; frequency of food intake; views on food and nutrition; and lifestyle, demographic and socioeconomic data [22]. The nationwide PolSenior2 survey conducted between 2017 and 2020 indicated that the prevalence of metabolic diseases, such as obesity, hypertension and type 2 diabetes, among the elderly was not significantly different in the Świętokrzyskie and Śląskie/Dolnośląskie regions. The lack of variation became the reason for selecting these regions to study the association between diet quality, demographic and socioeconomic characteristics and metabolic diseases [46]. The survey was conducted between the beginning of June and the end of September 2019. The study sample was chosen arbitrarily. A request was sent to all senior organizations (associations, foundations, clubs and senior circles) acting in the Świętokrzyskie region and Śląskie/Dolnośląskie region for consent to participate in the study. Additionally, the snowball method was applied in order to diversify the research sample of participants. Participants included in the study had to be at least 60 years old, represent one household and be able to complete the survey. A total of 750 questionnaires were distributed to sixteen senior organizations in both regions. A detailed survey sampling scheme is shown in Figure 1. 

The study was performed following the Declaration of Helsinki [47]. The respondents gave their written consent to participate in the study. Based on the provisions of the General Regulation of the European Parliament on Personal Data Protection, the personal data of the respondents were secured (GDPR 679/2016).

### 2.2. Dietary Data

Using the KomPAN questionnaire, the consumption of 24 food groups was assessed [22]. Habitual consumption over the past 12 months of each food group was assessed based on the following response categories (frequency of consumption): never (answer rank—(1)), several times a month (2), once a week (3), several times a week (4), once a day (5) and several times a day (6). Using the procedure for developing the data of the KomPAN questionnaire [22], two indices were calculated. To calculate them, the response ranks were converted into “semi-quantitative” data describing the frequency of daily consumption, which ranged from 0 to 2 times per day. On this basis, the “Pro-Healthy Diet Index” (pHDI-10) was extracted. It considers the potential health benefits relating to 10 food groups: (1) wholemeal bread and wholemeal bread rolls; (2) buckwheat, oats, wholegrain pasta or other coarse-ground groats; (3) milk; (4) fermented milk beverages (e.g., yogurts and kefir); (5) fresh cheese curd products (e.g., cottage cheese, homogenized cheese, fromage frais); (6) white meat (e.g., chicken, turkey, rabbit); (7) fish; (8) pulse-based foods (e.g., from beans, peas, soybeans, lentils); (9) fruit; and (10) vegetables. In addition, the “Unhealthy Diet Index” (nHDI-14), which represents the potential adverse health effect relating to 14 food groups, was derived: (1) white bread and bakery products (e.g., wheat bread, toast bread and white bread rolls); (2) white rice, white pasta and fine-ground groats (e.g., semolina and couscous); (3) fast foods; (4) fried foods (e.g., meat or flour-based foods, such as dumplings and pancakes); (5) butter (as a bread spread or as an addition to meals for frying, baking, etc.); (6) lard; (7) cheese (including processed cheese and blue cheese); (8) cold meats, smoked sausages and hot-dogs; (9) red meat (e.g., pork, beef, veal, mutton, lamb and game); (10) sweets; (11) tinned meats; (12) sweetened carbonated or still beverages; (13) energy drinks; and (14) alcoholic beverages [21]. 

In order to standardize the range of the two indices (pHDI and nHDI) and to facilitate their interpretation, the frequency of consumption (times/day) of products assigned to the pHDI index and the nHDI index were summed and expressed on a scale from 0 to 100 points [22]. The formulas used for the calculations were as follows: Pro-Healthy Diet Index (pHDI, in points) = (100/20) × sum of the frequency of the consumption of 10 food groups(times/day)
Unhealthy Diet Index (pHDI, in points) = (100/28) × sum of the frequency of the consumption of 14 food groups(times/day)
A range from 0 to 33 points for the index pHDI-10/nHDI-14 means low adherence to a healthy/unhealthy diet, 34–66 points indicate moderate adherence to a healthy/unhealthy diet and 67–100 points indicate high adherence to a healthy/unhealthy diet. As a result of the calculations carried out in accordance with the procedure [22], 5 indicators were obtained, i.e., 3 pHDI-10 indicators of three intensities of adherence to a healthy diet—namely, low, moderate and high—and 2 nHDI-14 indicators of two intensities of adherence to an unhealthy diet—namely, low and moderate. Since the participants’ diets were characterized by different frequencies of consumption of products included in the indicator pHDI-10 and nHDI- 14, in order to identify a specific structure of the quality of the diet of these people, 6 diet quality index profiles were developed. They were made up of combinations of the 5 indicators (3 pHDI-10 × 2 nHDI-14); see Table 1. 

Due to the small share of participants characterized by profiles numbered 1–3, they were excluded and, finally, 417 persons characterized by the three other diet quality index profiles numbered 4–6 qualified for analysis (Table 1). The profile numbered 4, which was characterized by indicators with the least potential health benefits, was named the “Lower Diet Quality Index”—L_DQI (moderate pHDI-10 and moderate nHDI-14). The profile numbered 6, which was characterized by indicators with a potential indirect health impact benefit, was called the “Middle Diet Quality Index”—M_DQI (high pHDI-10 and moderate nHDI-14). The profile with the number 5, which was characterized by the indicators with the most potential health benefits, was named the “Upper Diet Quality Index”—U_DQI (high pHDI-10 and low nHDI-10). 

### 2.3. Metabolic Diseases Data

To identify metabolic diseases, the questionnaire asked participants about their body weights (prevalence of obesity) and the prevalence of hypertension and type 2 diabetes (DMt2). In order to verify the presence of obesity in the surveyed elderly, the body mass index (BMI) was calculated based on the declared body mass and height. The data was adjusted based on previously developed regression equations for the elderly to be close to real (measured) values. [48]. The presence of obesity in the participants was considered to be a BMI of ≥30 kg/m^2^ [49,50]. 

### 2.4. Demographic and Socioeconomic Data

The demographic characteristics of the participants were assessed by considering the following factors: gender, age (in the ranges: 60–74 years and 75 years and more) and place of residence (village, city with less than 100,000 inhabitants and city with more than 100,000 inhabitants).

All the respondents answered the questions about a subjective assessment of socioeconomic status (SES). The questions concerned the following: the respondent’s financial situation (question 1), financial support from one’s family (question 2), financial social support (question 3) and education level (question 4). Points were assigned for each question response. The first question assessed the respondent’s financial situation based on the response categories: below average—1 point, average—2 points and above average—3 points. This question also assessed the economic situation of the respondent’s household based on the possible answers: I need to save to meet my basic needs—1 point; enough for my needs, but I need to save for larger purchases—2 points; and enough for me without saving—3 points. The second question assessed the financial assistance provided to the respondent by the family: 1 point was assigned for the answer “no, although I have financial problems”; 2 points for the answer “yes, because I have financial problems”; 3 points for the answer “no need because my financial situation is satisfactory”; and 4 points for the answer “yes, although I have no financial problems”. The third question concerned the financial assistance provided to the respondent by social institutions, with the same response categories as in question 2. The fourth question dealt with education with the following response categories: primary—1point, vocational—2 points, secondary—3 points and higher education—4 points.

Based on the SES index procedure [51,52], scores were summed for responses assessing the respondent’s socioeconomic status, and then respondents with low (L_SES), medium (M_SES) and high (H_SES) indices were separated using the tertiary distribution. In addition, Cronbach’s alpha index [53] was used to assess the reliability of the input of this index. The index value was 0.693.

### 2.5. Statistical Analysis

The analyses used the following qualitative variables (categorical values): gender (female, male), age (60–74, 75 and over), place of residence (rural, city < 100,000 inhabitants, city > 100,000 inhabitants), SES index (low, moderate, high) and diet quality index profiles (L_DQI, M_DQI, U_DQ). These variables were analyzed separately for each metabolic disease assessed (obesity, hypertension, type 2 diabetes). Qualitative variables were presented as numbers (*N*) and percentages (%). The chi-square test was used to verify the differences between these variables. A previously developed procedure was used to calculate the diet quality indicators [22]. The reliability of the data included in these indicators was confirmed using the Kaiser–Mayer–Olkin (KMO) measure of sampling adequacy and Bartlett’s test of sphericity. Statistical significance was achieved in both cases. The value of the KMO measure was 0.813. Bartlett’s test showed a significance of *p* < 0.0001. Based on the indicators, the three diet quality profiles were distinguished: L_DQI (Lower Diet Quality Index), M_DQI (Middle Diet Quality Index) and U_DQI (Upper Diet Quality Index). 

A logistic regression analysis was used to evaluate the relationship between the identified diet quality profiles (the DQIs), demographic characteristics and socioeconomic status (SES). The odds ratio (OR) value was calculated at the 95% confidence level. The reference group (OR = 1.00) consisted of L_DQI (Lower Diet Quality Index) and all categories of demographic characteristics and the SES index. A *p*-value less than 0.05 was considered significant for all tests. 

Statistical analysis was performed using STATISTICA statistical software (version 13.3 PL; StatSoft Inc., Tulsa, OK, USA; StatSoft, Kraków, Poland).

## 3. Results

### 3.1. Characteristics of the Study Sample

Table 2 shows the demographic and socioeconomic characteristics of the study group. A total of 417 people qualified for the study. The majority of the study group was represented by women, people aged 60–74 years, rural residents and those with a moderate socioeconomic status. 

### 3.2. Association of the Diet Quality with Declared Metabolic Diseases, Selected Demographic Characteristics and Socioeconomic Status

A variation in the diet quality among the study group according to the presence of declared metabolic diseases, selected demographic characteristics and socioeconomic status is presented in Table 3. 

Nearly three-quarters of the participants were characterized by the M_DQI profile and slightly less than one-sixth each was characterized by the L_DQI or U_DQI profile. 

The presence of obesity was declared by slightly more than a quarter of the surveyed group. Among these individuals, the L_DQI profile involved a significantly higher percentage of men, rural residents, and those with an L_SES or M_SES. The M_DQI profile mostly involved people aged 75 years or more and those with an H_SES. The highest prevalence of the U_DQI profile among people with obesity was among women, people aged 60–74 years and residents of small cities. 

The occurrence of arterial hypertension was declared by almost one-third of the surveyed group. Among these individuals, the L_DQI profile was predominant in males, people aged 75 years or more and rural residents. In turn, the M_DQI profile was mostly found in women, people aged 60–74 years, and those with an L_SES or H_SES. The highest significance of the U_DQI profile among arterial hypertension individuals was observed in urban residents and those characterized by M_SES.

The occurrence of diabetes type 2 was declared by one-sixth of the study group. Among these people, the L_DQI profile was mostly found in men, people aged 60–74 years, rural residents and people with an M_SES. The M_DQI profile was mostly found in women, residents of small towns and people with an L_SES. The U_DQI profile among people with DMt2 was found to be most significant in those who were aged 75 years or older, were residents of large cities and with an H_SES.

### 3.3. Measuring the Strength of the Association between Diet Quality and Declared Metabolic Diseases, Selected Demographic Characteristics and Socioeconomic Status

The results of the logistic regression are shown in Table 4. They showed that among the participants declaring a presence of obesity, the M_DQI profile in relation to the L_DQI profile was significantly more common in subjects living in large cities compared with those from rural areas (OR = 1.66), and subjects with an H_SES compared with those with an L_SES (OR = 1.71). While the U_DQI profile in relation to the L_DQI profile was significantly more often in women than in men (OR = 2.83), more often in residents of small towns than villages (OR = 2.25) and more often in residents of big towns than villages (OR = 1.67). For those declaring a presence of arterial hypertension, the M_DQI profile in relation to the L_DQI profile was significantly more common in women than in men (OR = 1.67) and for those aged 60–74 years than for those aged 75 years and more (OR = 2.05). On the other hand, the U_DQI profile in relation to the L_DQI profile was found significantly more often in residents of small towns than villages (OR = 1.88) and more often in residents of big towns than villages (OR = 2.06) and people with an M_SES than with an L_SES (OR = 1.94). Among subjects declaring type 2 diabetes mellitus, the M_DQI profile in relation to the L_DQI profile was significantly more common in women than in men (OR = 2.24), residents of small towns than those of rural areas (OR = 2.82) and those with an L_SES than those with an H_SES (OR = 1.64). On the other hand, the U_DQI profile in relation to the L_DQI profile was significantly more frequent for those aged 75 years or more than for those aged 60–74 years (OR = 1.93), residents of large cities than rural areas (OR = 2.98), those with an H_SES than an L_SES (OR = 1.86) and with an H_SES than an M_SES (OR = 2.05).

## 4. Discussion

The present study assessed Polish seniors to establish the relationship between the quality of diet and the prevalence of obesity, arterial hypertension and diabetes type 2, as well as demographic characteristics and socioeconomic status. This is a complex and ambiguous issue and, to our knowledge, has not been investigated by other researchers in such a broad manner.

According to the Polish Central Statistical Office (GUS), in Poland in 2019, obesity (BMI ≥ 30.0 kg/m^2^) among older women (≥60 years) ranged from 21.8% for the age group ≥ 80 years to 30.0% for the persons aged 70–79 years. In the case of men, it was 19.3% and 27.4%, respectively [54]. It can be assumed that the percentage of obesity (26.4%) based on the declarations of our participants was in the range reported by the GUS. 

Diet quality seems to influence obesity risk and obesity more than the relative amount of macronutrients in the consumed food [54,55,56,57]. However, only a few prospective studies investigated the association of diet quality with obesity risk [56,58,59]. Most of them found that a higher-quality diet, such as, e.g., the traditional Mediterranean diet, was inversely associated with the risk of obesity or weight gain [56,58]. An inverse relationship between the Mediterranean diet and obesity rates was also reported in many [39,60,61,62] cross-sectional studies, but not in all [63]. 

Although the present study had the highest percentage of obese participants in the moderate diet quality index profile (68.2%), those characterized by the lower profile were more than twice as many as those with the upper diet quality index profile (21.8% vs. 10.0%), which may suggest a greater association of low diet quality with obesity.

Nowadays, there is an emphasis put on studying the impact of dietary and food qualities on health and their association with sociodemographic factors. However, the most common studies are those performed in populations of children, teenagers and young adults [64,65,66]. There is a lack of such research regarding older adults. The present study showed that obesity was more often associated with poor diet quality among older men living in rural areas with a low or moderate SES. In addition, there is a lack of investigations that combines all of the abovementioned aspects. 

Zhu et al. showed that people with low socioeconomic status tend to lead unhealthy lifestyles, including eating foods with low nutritional value and limiting physical activity. The results confirmed that the intake of foods that benefit the quality of the diet (vegetables, fruits, wholegrain products, milk and fish) increased with the higher SES of overweight individuals [67]. Our research gave similar findings, meaning that a low-quality diet was significantly more common among participants with an L_SES and M_SES. This is most probably a consequence of the financial resources. Drewnowski and Darmon proved that higher-energy-density food has lower costs and can be a good way to save money [68].

In turn, the study by López-Olmedo et al. led to different results, which showed that a low SES was positively correlated with higher Mexican diet quality indicators. This was most likely due to the higher consumption of legumes and wholegrain products, which are typical and popular in the diets of poorer people in Mexico [69]. 

The authors of the present study also showed that a low-quality diet is more common in obese men than in obese women (Table 2). Gómez et al. [70] showed, yet another trend, namely, the prevalence of overweight and obesity was higher among women, especially those with low SES. This difference might have been due to the level of education, as it is the important SES determinant. Lower education level is more common among people with low or middle earnings, which is often associated with unhealthy dietary choices. De Mello et al. correlated these indicators with the higher prevalence of obesity in developing countries [71]. In 2018, de Assumpção et al. conducted a study on Brazilian women and showed a higher intake of products that benefit diet quality among women with higher education. It is assumed that people with higher levels of education are more aware of the impact of different kinds of nutrition on health [72].

Significant differences in diet quality were observed among seniors living in rural and urban areas. Park et al. found that the consumption of vegetables, fruits and protein sources was significantly lower among rural residents compared with urban inhabitants [73]. Our study also showed that a low-quality diet was more common in seniors living in rural areas (Table 2). This might be caused by limited access to high-quality food among the population living in such areas [74]. It was shown that the accessibility of healthy food stores and product prices were significantly associated with the consumption of high-quality food [74,75,76]. In addition, those people mainly use food products from their farms, which may affect the limited food diversity [77]. However, the study by Park et al. [73] found that chronic diseases, including obesity, affect rural residents less often compared with urban residents. This might be due to the limited access to medical services in smaller towns and cities, and thus, the subsequent diagnosis of diseases among the elderly.

Our research showed that a high-quality diet was significantly more common among residents of small (OR = 2.25) and large cities (OR = 1.67) compared with those living in rural areas (Table 3). This can be affected by the fact that urban residents, as those who are more aware, more often practice and promote healthy eating habits. In addition, they usually have easier access to a variety of programs supporting seniors’ health [78]. Another study suggested that the reason for the lower quality diet of seniors living in small towns is the long distance to grocery stores, which makes it difficult to purchase and regularly consume products, especially unprocessed ones [79]. 

According to the latest 2019 Report of the Polish NHF, 31.5% of the Polish adult population suffered from arterial hypertension in 2018, and the percentage increased with age, achieving, depending on the age group, 74% to 84% of older women and 67% to 77% of older men [80].

In the present self-reported survey, the percentage of elderly people declaring the presence of arterial hypertension was much lower than that reported by the Polish NHF and amounted to 31.2% (Table 2). Such large differences are most likely due to the selection of participants in our study. Respondents in our study were active seniors that participated in various associations, foundations, clubs, etc. According to the results of other authors, older people who are engaged in various types of social activities feel satisfied with their lives, and their quality of life is stable and kept at a high level. Such people also have better access to life-long education on health and willingly use it, which influences their dietary choices [81]. 

Some indicators of a low diet quality are associated with the risk of arterial hypertension, especially a significant salt intake [82]. Arterial hypertension in our study was found to be substantially related to a low-quality diet in men aged ≥75 years living in rural areas. Similar to the case of obesity, there is a lack of research results that take into account all the factors previously mentioned. Some findings are consistent with our results, as they show that low SES is more often associated with unhealthy diet features. This is explained by the lack of nutritional knowledge and the reluctance to acquire it among people of lower socioeconomic status, which is largely reflected in their dietary choices [83]. 

The present study showed that arterial hypertension was significantly more common in elderly men than in elderly women (OR = 1.67, *p* = 0.044) (Table 3). Different results were obtained in the study by Chinnakali et al. The significantly higher occurrence of arterial hypertension among women was probably due to their higher anthropometric indices compared with men, i.e., mean BMI 26.84 ± 4.86 kg/m^2^ vs. 24.19 ± 4.42 kg/m^2^ and WHR 94.88 ± 6.19 vs. 91.54 ± 6.33, respectively [84]. 

It was suggested that using the distinguishing features of a high-quality diet can help lower blood pressure in people suffering from hypertension [85,86]. However, Daneshzad et al. [87] and Motamedi et al. [88] found no association between the HEI (Healthy Eating Index) and both systolic and diastolic blood pressure values. Discrepancies in the study results could have been caused by the different sample sizes, health statuses of the participants, possibility of confounding factors and choice of methods used to assess food intake. 

The authors of the present study also showed that arterial hypertension in rural residents was more often associated with a low-quality diet. The results obtained by Chantakeeree et al. confirmed that the higher quality of life of seniors living in cities was associated with higher economic status and family support [89]. While there are results of Chinese studies available that show that arterial hypertension affected urban residents more often, in the quoted studies, no connection to the diet was considered [90,91]. The higher susceptibility to arterial hypertension of urban residents might be explained by the higher intensity of work pressures and interpersonal relationships, as well the more frequent consumption of the unhealthy foods more commonly available in cities. The second important factor that can affect the higher percentage of arterial hypertension in city inhabitants is easier access to health care, and thus, faster and more frequent diagnosis of the disease. 

Diabetes is also a serious public health problem [92]. According to the recent expert report prepared by the National Institute of Public Health–Polish Institute of Hygiene in 2019, the prevalence of diabetes was 32.6% among the elderly aged 65–74 years and 24.8% among those aged 75 years or more. A slightly higher share of people with type 2 diabetes mellitus were women [93]. In our study, the percentage of people with diabetes was lower and amounted to 16.1%. 

Diet is a major factor in modifying the incidence and course of diabetes mellitus type 2 [94]. Observational, prospective and clinical studies demonstrated healthy diet importance in the prevention and treatment of diabetes [95]. Examples include the Mediterranean diet [36] and DASH diet [96]. In addition, a meta-analysis showed that the Mediterranean diet, DASH diet and diet quality, as assessed using the Alternate Healthy Eating Index (AHEI), were closely correlated with a reduced risk of diabetes, even in the presence of various specific dietary components or products [97]. In our study, a higher proportion of people with diabetes type 2 concerned the middle and upper profile of diet quality, which should be explained by the use of higher diet quality when diabetes is present.

Several studies showed that a higher-quality diet, based on adherence to the Mediterranean diet, was associated with a lower incidence of obesity, arterial hypertension or diabetes mellitus type 2 [38,98], but also had a positive effect on metabolic parameters that determine the incidence of these diseases, namely, BMI, WC and WHR values, as well as blood levels of triglycerides, total cholesterol, LDL-cholesterol, HDL-cholesterol or fasting glucose [38,99]. As in the case of obesity and arterial hypertension, diabetes type 2 was also more common among male seniors living in rural areas whose diets were of low quality. The results published by other authors also indicated that the quality of women’s diets is higher than that of men’s [100,101]. Such results are explained by women’s greater concern for their health and that of their families [102]. Women were also shown to have higher levels of awareness related to diabetes. [103]. Furthermore, higher education and a concurrent high SES were significantly associated with higher diet quality [104]. In a study of Brazilian seniors, it was observed that those with higher education consumed foods that were characterized by a lower carbohydrate content but rich in minerals and vitamins compared with those who did not complete primary school [105]. It was also shown that elderly people diagnosed with diabetes had a higher diet quality. This was most likely due to the fact that the diagnosis of the disease contributed to the increased motivation associated with changing eating habits. In addition, the use of properly balanced rations with a low glycemic index is the cornerstone of diabetes type 2 treatment, and thus, changes in diet are almost mandatory for diabetics [106]. Moreover, it was noted that elderly people with diabetes or with a pre-diabetic condition who live in rural areas most often do not pay attention to glycemic control and consume products rich in sugars, fat and salt [107].

The dangers of health risk factor impacts change with age and vary at different periods of life. Among the elderly, health risk factors include low physical activity, high blood pressure, low consumption of fruits and vegetable, high cholesterol, overweight and obesity, and smoking. Similar factors are observed at younger ages. This shows the relative persistence of these factors in their conditions and long-term effects on the body. This is particularly important in relation to metabolic diseases, for which the timing of exposure to risk factors can be critical in the development of these diseases, their course and mortality [27]. 

The hitherto research is more likely to show the prevalence of more metabolic disease risk factors among rural residents than urban residents, and among those with a low socioeconomic status [108,109,110,111]. Discrepancies regarding the association between the diet quality indicators and socioeconomic factors with parameters associated with metabolic diseases might also be due to differences in genetic background, gene–environment interactions and the presence of confounding factors.

### Strengths and Limitations of the Study

The results obtained in the present study can contribute to the development of the actions undertaken by public health professionalsaimed at changing dietary behavior and improving the quality of seniors’ diets, especially those diagnosed with metabolic diseases. In addition, to our knowledge, this is the very first study that broadly defines the association between the quality of the elderly persons’ diet; the prevalence of obesity, arterial hypertension and diabetes type 2; demographic characteristics; and socioeconomic status. However, this study had some limitations. This was a cross-sectional study; therefore, it was not possible to establish a cause-and-effect relationship between the variables and evaluate changes over time. Furthermore, the survey was not conducted on a representative sample; respondents were recruited from only two regions of Poland. In addition, it was a group of socially active seniors. In addition, due to the low response of senior organizations and the dominance of women in these organizations, the research sample became heterogeneous. This can negatively affect the possibility of a correct inference. Therefore, it is not possible to put forward this study’s results as general results for the whole Polish population, as well as for other countries, due to cultural and economic differences, among other things. Although cross-sectional studies provide valuable information on the association of diet quality with metabolic diseases and demographic and socioeconomic characteristics, longitudinal studies are necessary to obtain stronger findings in this field.

## 5. Conclusions

Independent of the factors that determine the quality of diet, it was concluded that even small changes toward a healthier diet can contribute to a reduction in the risk of metabolic diseases [37], longer life and better quality of life [112]. However, there is insufficient evidence demonstrating the simultaneous association of metabolic diseases with diet quality and demographic and socioeconomic factors. Although the present study was characterized by factors that limit the strength of the association between these variables, it gives grounds to conclude that population characteristics, such as male gender, rural residence and lower socioeconomic status, may be associated with lower diet quality, while female gender, urban residence and higher socioeconomic status are associated with higher diet quality among elderly people with metabolic diseases. The relationship between diet quality, demographic characteristics and socioeconomic status was demonstrated, but it was not possible to obtain unambiguous results on the relationships of these variables with the occurrence of metabolic diseases. Future extended studies should appraise the importance of diet quality in reducing the risk of metabolic diseases among the elderly, taking into account variability due to the environmental characteristics of the study population. 

## Figures and Tables

**Figure 1 ijerph-20-02878-f001:**
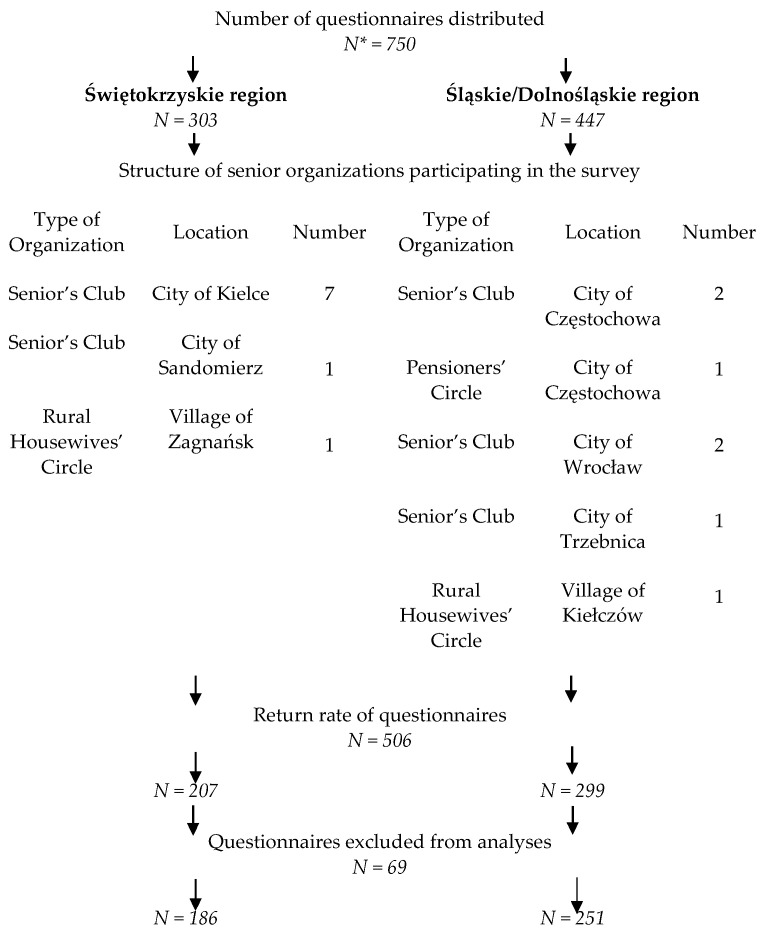
Research sampling scheme. * *N*—number of questionnaires.

**Table 1 ijerph-20-02878-t001:** Diet quality index profiles with the proportion of study participants characterized by a given profile.

Profile Number	Type of Indicator	Participation of Participants (*N* * = 437)	
		*N*	%
1	Low pHDI-10 and low nHDI-14	2	0.5
2	Low pHDI-10 and moderate nHDI-14	9	2.1
3	Moderate pHDI-10 and low nHDI-14	9	2.1
4	Moderate pHDI-10 and moderate nHDI-14	61	14.0
5	High pHDI-10 and low nHDI-10	55	12.5
6	High pHDI-10 and moderate nHDI-14	301	68.8

* *N*—number of participants.

**Table 2 ijerph-20-02878-t002:** Characteristics of the study sample.

Characteristic	Total	
	*N* *	%
Total	417	100.0
Gender		
Women	275	65.9
Men	142	34.1
Age		
60–74 years old	318	76.3
75 years and more	99	23.7
Place of residence		
Village	194	46.5
City < 100,000 inhabitants	86	20.6
City > 100,000 inhabitants	137	32.9
Index SES		
L_SES	128	30.7
M_SES	177	42.4
H_SES	112	26.9

* *N*—number of participants.

**Table 3 ijerph-20-02878-t003:** Differentiation of diet quality according to the presence of declared metabolic diseases, selected demographic characteristics and socioeconomic status.

Metabolic Diseases	Demographic and Socioeconomic Features	Total		Diet Quality Index Profiles					
				L_DQI ^a^		M_DQI ^b^		U_DQI ^c^	
		*N*	%	*N*	%	*N*	%	*N*	%
	Total	417	100.0	61	14.6	301	72.2	55	13.2
Obesity	Total ^ab,bc,ac^	110	26.4	24	21.8	75	68.2	11	10.0
	Gender								
	Women ^ab,bc,ac^	72	65.5	14	58.3	49	65.3	9	81.8
	Men ^ab,bc,ac^	38	34.5	10	41.7	26	34.7	2	18.2
	Age								
	60–74 years old ^ab,bc,ac^	82	74.5	18	75.0	54	72.0	10	90.9
	75 years or over ^ab,bc,ac^	28	25.5	6	25.0	21	28.0	1	9.1
	Place of residence								
	Village ^ab,bc,ac^	56	50.9	17	70.8	35	46.7	4	36.4
	City < 100,000 inhabitants ^ab,bc,ac^	24	20.8	2	8.3	18	24.0	4	36.4
	City > 100,000 inhabitants ^ab,bc,ac^	30	27.3	5	20.9	22	29.3	3	27.2
	SES Index								
	L_SES ^ab,bc^	37	33.6	11	45.8	22	29.3	4	36.4
	M_SES ^ab,bc,ac^	49	44.6	12	50.0	32	42.7	5	45.4
	H_SES ^ab,bc,ac^	24	21.8	1	4.2	21	28.0	2	18.2
Total ^ab,bc,ac^	Total ^ab,bc,ac^	130	31.2	26	20.0	86	66.2	18	13.8
	Gender								
	Women ^ab,bc,ac^	83	63.8	13	50.0	58	67.4	12	66.7
	Men ^ab,bc^	47	36.2	13	50.0	28	32.6	6	33.3
	Age								
	60–74 years old ^ab,bc.ac^	94	72.3	19	73.1	65	75.6	10	55.6
	75 years or over ^ab,bc^	36	27.7	7	26.9	21	24.4	8	44.4
	Place of residence								
	Village ^ab,bc,ac^	60	46.2	13	50.0	41	47.7	6	33.3
	City < 100,000 inhabitants ^ab,bc,ac^	24	18.5	5	19.2	15	17.4	4	22.2
	City > 100,000 inhabitants ^ab,bc,ac^	46	35.3	8	30.8	30	34.9	8	44.5
	SES Index								
	L_SES ^bc^	60	46.2	12	46.2	41	47.7	7	38.9
	M_SES ^ab,bc,ac^	47	36.2	11	42.3	28	32.6	8	44.4
	H_SES ^ab,bc,ac^	23	17.6	3	11.5	17	19.7	3	16.7
Total ^ab,bc,ac^	Total ^ab,bc,ac^	67	16.1	9	13.4	44	65.7	14	20.9
	Gender								
	Women ^ab,bc,ac^	43	64.2	4	44.4	30	68.2	9	64.3
	Men ^ab,bc^	24	35.8	5	55.6	14	31.8	5	35.7
	Age								
	60–74 years old ^ab,bc,ac^	49	73.1	7	77.8	33	75.0	9	64.3
	75 years and over ^ab,bc,ac^	18	26.9	2	22.2	11	25.0	5	35.7
	Place of residence								
	Village ^ab,bc,ac^	28	41.8	5	55.6	18	40.9	5	35.7
	City < 100,000 inhabitants ^ab,bc,ac^	16	23.9	1	11.1	12	27.3	3	21.4
	City > 100,000 inhabitants ^ab,bc,ac^	23	34.3	3	33.3	14	31.8	6	42.9
	SES Index								
	L_SES ^ab,bc,ac^	27	40.3	3	33.3	19	43.2	5	35.7
	M_SES ^ab,bc,ac^	23	34.3	5	55.6	13	29.5	5	35.7
	H_SES ^ab,bc,ac^	17	25.4	1	11.1	12	27.3	4	28.6

^ab, bc, ac^ significant differences between diet quality index profiles with demographic and socioeconomic characteristics; chi-square test; *p* < 0.05.

**Table 4 ijerph-20-02878-t004:** Odds ratios for moderate- and high-diet-quality indexes for selected demographic and socioeconomic characteristics.

Demographic and Socioeconomic Features	Metabolic Diseases											
	Obesity				Arterial Hypertension				Diabetes Mellitus Type 2			
	Diet Quality Index Profiles (Ref. ^a^ L_DQI)											
	M_DQI		U_DQI		M_DQI		U_DQI		M_DQI		U_DQI	
	OR ^b^	*p*	OR	*p*	OR	*p*	OR	*p*	OR	*p*	OR	*p*
Gender												
Women (ref. ^a^)	1.00		1.00		1.00		1.00		1.00		1.00	
Men	0.73 (0.40–1.25)	0.218	**0.28** (0.20–0.47)	**<0.001**	**0.44** (0.31–0.73)	**0.045**	0.73 (0.47–1.12)	0.291	**0.45** (0.28–0.72)	**<0.001**	0.71 (0.41–1.23)	0.214
Men (ref.)	1.00		1.00		1.00		1.00		1.00		1.00	
Women	1.45 (0.80–2.48)	0.218	**2.83** (1.50–4.21)	**<0.001**	**1.67** (1.01–2.77)	**0.044**	1.35 (0.77–2.37)	0.291	**2.24** (1.41–3.58)	**<0.001**	1.42 (0.82–2.46)	0.213
Age												
60–74 years old (ref.)	1.00		1.00		1.00		1.00		1.00		1.00	
75 years or over	1.39 (0.84–2.18)	0.337	0.59 (0.34–0.99)	0.055	**0.50** (0.31–0.82)	**0.004**	0.66 (0.38–1.16)	0.132	1.20 (0.75–1.93)	0.438	**1.93** (1.18–3.17)	**0.008**
75 years or over (ref.)	1.00		1.00		1.00		1.00		1.00		1.00	
60–74 years old	0.71 (0.45–1.21)	0.338	1.60 (1.01–2.64)	0.056	**2.05** (1.25–3.36)	**0.004**	1.54 (0.88–2.69)	0.132	0.83 (0.52–1.33)	0.438	**0.52** (0.32–0.85)	**0.008**
Place of residence												
Village (ref.)	1.00		1.00		1.00		1.00		1.00		1.00	
City < 100,000	1.31 (0.78–2.09)	0.317	**2.25** (1.37–3.62)	**0.001**	1.48 (0.86–2.55)	0.151	**1.88** (1.34–2.64)	**<0.001**	**2.82** (1.71–4.53)	**<0.001**	1.58 (0.93–2.69)	0.086
City > 100,000	**1.66** (1.00–2.79)	**0.049**	**1.67** (1.04–2.75)	**0.030**	1.36 (0.88–2.09)	0.168	**2.06** (1.47–2.88)	**<0.001**	1.50 (0.87–2.60)	0.145	**2.98** (1.92–4.88)	**<0.001**
City < 100,000 (ref.)	1.00		1.00		1.00		1.00		1.00		1.00	
City > 100,000	1.36 (0.84–1.16)	0.213	0.80 (0.51–1.25)	0.321	0.96 (0.56–1.64)	0.876	1.21 (0.70–2.10)	0.485	0.79 (0.46–1.36)	0.308	1.36 (0.85–2.19)	0.195
Village	0.77 (0.48–1.25)	0.327	**0.59** (0.38–0.92)	**0.018**	0.67 (0.39–1.16)	0.151	**0.48** (0.27–0.83)	**<0.001**	**0.33** (0.20–054)	**<0.001**	0.63 (0.37–1.07)	0.087
City > 100,000 (ref.)	1.00		1.00		1.00		1.00		1.00		1.00	
Village	**0.59** (0.38–0.99)	**0.049**	0.69 (0.44–1.09)	0.109	0.74 (0.48–1.14)	0.168	**0.45** (0.28–0.73)	**<0.001**	0.67 (0.39–1.15)	0.145	**0.27** (0.19–0.39)	**<0.001**
City < 100,000	0.73 (0.45–1.18)	0.220	1.25 (0.80–1.95)	0.321	1.04 (0.61–1.78)	0.876	0.79 (0.49–1.25)	0.485	1.27 (0.80–2.02)	0.308	0.73 (0.47–1.14)	0.195
Index SES												
L_SES (ref.)	1.00		1.00		1.00		1.00		1.00		1.00	
M_SES	1.35 (0.84–2.15)	0.213	1.09 (0.69–1.72)	0.724	1.14 (0.72–1.80)	0.576	**1.94** (1.20–3.15)	**0.007**	0.75 (0.47–1.19)	0.217	0.98 (0.62–1.53)	0.928
H_SES	**1.71** (1.10–2.67	**0.018**	1.35 (0.79–2.31)	0.268	1.01 (0.64–1.61)	0.955	1.21 (0.70–2.10)	0.485	**0.61** (0.37–1.00)	0.049	**1.86** (1.14–3.02)	**0.012**
M_SES (ref.)	1.00		1.00		1.00		1.00		1.00		1.00	
H_SES	1.30 (0.82–2.07)	0.260	1.24 (0.71–2.15)	0.449	0.85 (0.54–1.34)	0.484	0.63 (0.36–1.11)	0.107	0.77 (0.48–1.25)	0.326	**2.05** (1.25–3.36)	**0.004**
L_SES	0.75 (0.47–1.19)	0.217	0.92 (0.58–1.46)	0.724	0.88 (0.55–1.39)	0.576	**0.52** (0.32–0.84)	**0.007**	1.33 (0.83–2.13)	0.214	1.03 (0.66–1.61)	0.904
H_SES (ref.)	1.00		1.00		1.00		1.00		1.00		1.00	
L_SES	**0.59** (0.38–0.92)	**0.018**	0.74 (0.43–1.26)	0.268	0.99 (0.60–1.62)	0.958	0.83 (0.48–1.44)	0.501	**1.64** (1.00–2.69)	**0.049**	**0.54** (0.32–0.87)	**0.012**
M_SES	0.77 (0.48–1.22)	0.260	0.81 (0.47–1.40)	0.449	1.18 (0.75–1.85)	0.485	1.59 (0.90–2.79)	0.106	1.28 (0.79–2.07)	0.317	**0.50** (0.31–0.82)	**0.005**

^a^ Reference group; ^b^ point estimate at 95% Wald confidence; *p*—significance level of the Wald’s test. Significant odds ratios are bolded.

## Data Availability

Data presented in this study are available on request from the corresponding author.

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
