# Peer review of "Diet Quality Variation among Polish Older Adults: Association with Selected Metabolic Diseases, Demographic Characteristics and Socioeconomic Status"

_ijerph, 2023, doi:10.3390/ijerph20042878_

Round 1

Reviewer 1 Report

Comments and revision

1. It is suggested to remove the type of study from the title.

2. Abstract: The abstract does not specify the type of study and the selection criteria. Confusing terms such as association, relationship, incidence, risk is mentioned, which is suggested to be reviewed and standardized according to the study. The conclusions are not new or clear. It is suggested to review the wording and congruence of the abstract.

3. The antecedents addressed the limitations of the instruments to assess the quality of the diet, which, although important, does not correspond to the main objective, in addition to continuing with the types of diets that have been proposed for cardiovascular health such as the DASH diet, the Mediterranean diet among others, which continues to be unrelated or unrelated to the title and objective of the study. Mention is made of several problems of the adult over 60 years of age, but they are data that are already known, including causal factors such as the type and quality of the diet, mentioning non-communicable diseases to cognitive functions. Therefore, the introduction should focus on the main topic of the study, on what is and is not in the literature.

4. Mention the study design.

5. Specify the selection criteria promptly, it is recommended to make an algorithm or graph that shows the recruitment procedures of each place.

6. Although GDP is mentioned as a criterion for cities, it is recommended to make a sociodemographic description of each selected province and justify why these and not others. Example, non-communicable disease statistics, occupation, demographics, etc.

7. In lines 25 to 119, it is mentioned that a questionnaire was applied, but it does not mention what this questionnaire consists of.

8. A frequency questionnaire is mentioned for the evaluation of the diet, but it is not mentioned how the memory issue was controlled, if there were caregivers or other relatives or people with the participants. For this reason, the inclusion and non-inclusion criteria are requested to be explained in detail.

9. Due to age, the population is susceptible to cognitive or senile problems, or typical of aging, and the study has surveys in all its methodology, so the authors are asked to specify how these variables are controlled.

10. In line 144 it is mentioned “In order to standardize the range of the both indices, the sum of frequency of food…” Was the standardization arbitrary or was it validated or is it already validated?

11.Why is the BMI used? It has limitations for older adults.

12.The procedures are confusing; the main objective, the main variables, the capture instruments are not mentioned; an attempt is made to mention a validation, but the design does not correspond to a validation; the perception of the participants is used and there is no clarity in the procedures and confidence in these procedures (Lines 160 to 175).

13. Statistical analysis qualitative variables are mentioned, but the type of variables is not mentioned nor are the objectives clear.

14.Table 2 shows heterogeneity in the sample studied to draw conclusions, but no mention is made of it. Twice the sample was made up of women and by age group it was 60 to 74, it is mentioned coming from the city and the province, which is not mentioned in the selection criteria.

15. For example, the education of the participants and the gynecological and obstetric history of the women are not mentioned, which are variables that must be controlled since this has or shows cardiovascular differences as well as the reliable diagnosis and not only referred.

16.It is suggested that the conclusion and discussion be made according to the "Strengths and Limitations of the Study" section, since it does not agree.

Final comments: Although it is a study of interest and that it is needed, the study lacks a methodological quality with the control and identification of biases, definition of the variables, as well as the design of the study and all its methodology. It is recommended to review these observations and suggestions, but above all, the congruence and consistency in the terms throughout the study.

Reviewer 2 Report

The author have postulated several hypothesis that remain largely unanswered due to the kind of statistical design and analyses. The introduction was apt, but several details in the Methods and Results were missing. 
